# Splenosis Mimicking Peritoneal Seeding of Advanced Colon Cancer Can Be Identified by Spleen SPECT/CT and FDG PET/CT

**DOI:** 10.3390/diagnostics11061045

**Published:** 2021-06-07

**Authors:** Ji Young Lee, Hee-Sung Song, Jimin Han

**Affiliations:** College of Medicine, Jeju National University, Jeju 63241, Korea; easy02000@naver.com (J.Y.L.); cocoro503@naver.com (J.H.)

**Keywords:** splenosis, peritoneal seeding, colon cancer, spleen SPECT/CT, FDG PET/CT

## Abstract

This case report demonstrates that Tc-99m labeled heat-damaged red blood cell single-photon emission computed tomography/computed tomography (spleen SPECT/CT) and F-18 fluorodeoxyglucose positron emission tomography/CT (FDG PET/CT) could noninvasively confirm splenosis mimicking peritoneal seeding of advanced sigmoid colon cancer with hepatic metastases, and played a crucial role in determining the treatment plan.



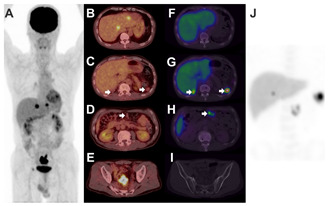



A 56-year-old man with sigmoid colon cancer was referred to the Department of Nuclear Medicine for the evaluation of multiple nodular lesions in the liver, and perihepatic, splenic fossa, and omental areas by enhanced abdominal CT. FDG PET/CT was performed ((**A**), maximum intensity projection image), which shows hypermetabolic lesions with intense FDG uptake in the liver (SUVmax 9.3, 6.7) and sigmoid colon (SUVmax 15.8), suggestive of colon cancer with hepatic metastases (**B**,**E**). There was no FDG uptake in other abdominal lesions, raising the suspicion of a benign lesion such as splenosis, considering his history of traumatic splenectomy before 10 years (**C**,**D** arrows). However, peritoneal seeding could not be excluded, considering the partial volume effects of FDG PET/CT and the aggravated clinical stage of the patient. Spleen SPECT/CT was performed ((**J**), maximum intensity projection image), which shows significant radioactivity in the perihepatic, splenic fossa, and omental areas (**G**,**H** arrows) and no radioactivity in the sigmoid colon tumor and hepatic lesions (**F**,**I**). The physicians excluded the possibility of peritoneal carcinomatosis and could provide the patient with treatment options, such as surgery. Splenosis was first described by Buchbinder and Lipkoff in 1939 as the heterotopic autotransplantation of splenic tissue mainly in the peritoneal cavity after traumatic spleen rupture [1,2]. Early diagnosis of splenosis is important to determine the treatment regimen because splenic implants may mimic peritoneal metastases in cancer patients. Spleen SPECT/CT using Tc-99m labeled heat-damaged red blood cells can confirm splenosis noninvasively [3]. FDG PET/CT is a useful modality to detect peritoneal seeding due to increased glucose metabolism of the lesions [4]. There are few references on spleen SPECT/CT and FDG PET/CT of splenosis in aggravated colon cancer patients. Physicians often worry about false-negative results of FDG PET/CT because of the partial volume effect on small lesions. Positive findings on spleen SPECT/CT can give them additional confidence in the results of FDG PET/CT and negative findings on the test can warn them that further examination is necessary such as pathologic correlations. The present case demonstrates that FDG PET/CT and spleen SPECT/CT can identify splenosis mimicking peritoneal seeding in patients with advanced colon cancer because splenosis is associated with low glucose metabolism and uptake of heat-damaged red blood cells.

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
