# Peer review of "Splenosis Mimicking Peritoneal Seeding of Advanced Colon Cancer Can Be Identified by Spleen SPECT/CT and FDG PET/CT"

_diagnostics, 2021, doi:10.3390/diagnostics11061045_

Round 1
Reviewer 1 Report
This is an interesting article presenting the image of the Spleen SPECT/CT. Nevertheless, routine FDG-PET/CT comprises satisfactory sensitivity to differentiate metastatic CRC disease from bening conditions such as splenosis. Therefore, the authors should document the clinical benefit and the impact on cost-effectiveness of an additional PET/CT.Author Response
We appreciate thoughtful comments. As you mentioned, FDG PET/CT can comprises satisfactory sensitivity to differentiate metastatic CRC disease from bening conditions. But, physicians often worry about false-negative results (small peritoneal seeding without FDG uptake) because the partial volume effect of FDG PET/CT on small lesions. Positive findings on spleen SPECT/CT can give doctors additional confidence in the results of FDG PET/CT and negative findings on the test can warn them that further examination is necessary such as pathologic correlations. Also, the cancer patients covered by Korea's national health insurance do not pay much on additional spleen SPECT/CT (about 30 USD).
We hope that this article answers your comments and we revised the manuscript based on your good advice (red words).

Reviewer 2 Report
The manuscript of case report entitled “ Splenosis Mimicking Peritoneal Seeding of Advanced Colon Cancer Can Be Identified by Spleen SPECT/CT and FDG PET/CT” provide a new approach that can non-invasively confirm splenosis mimicking peritoneal seeding of advanced sigmoid colon cancer with hepatic metas-tases, and played a crucial role in determining the treatment plan through using Tc-99m labeled heat-damaged red blood cell single photon emission computed tomography/computed tomography (spleen SPECT/CT) and F-18 fluorodeoxyglucose positron emission tomography/CT (FDG PET/CT). It will be useful in clinical treatment.
Author Response
We appreciate thoughtful and nice comments.
Reviewer 3 Report
The method is very interesting and if confirmed it can be of great help for the immediate treatment of the therapy.
The image is very clear and describes what happens well
Author Response

(The authors gave the same response as above.)

Round 2
Reviewer 1 Report
The manuscript has been improved after authors' modification.